# Biosynthesis of Pyrrole-2-carbaldehyde via Enzymatic CO$_2$ Fixation

Gabriel R. Titchiner [ID], Stephen A. Marshall [†] [ID], Herkus Miscikas and David Leys *

Manchester Institute of Biotechnology, University of Manchester, 131 Princess Street, Manchester M1 7DN, UK; gabriel.titchiner@manchester.ac.uk (G.R.T.); stephen.marshall@chem.ox.ac.uk (S.A.M.); herkusm@gmail.com (H.M.)
* Correspondence: david.leys@manchester.ac.uk
† Current address: Chemistry Research Laboratory, University of Oxford, Oxford OX1 3TA, UK.

**Abstract:** The use of CO$_2$ as a chemical building block is of considerable interest. To achieve carbon fixation at ambient conditions, (de)carboxylase enzymes offer an attractive route but frequently require elevated [CO$_2$] levels to yield the acid product. However, it has recently been shown that the coupling of a UbiD-type decarboxylase with carboxylic acid reductase yields the corresponding aldehyde product at near ambient [CO$_2$]. Here, we show this approach can be expanded to different UbiD and CAR enzymes to yield alternative products, in this case, the production of pyrrole-2-carbaldehyde from pyrrole, using *Pseudomonas aeruginosa* HudA/PA0254 in combination with *Segniliparus rotundus* CAR. This confirms the varied substrate range of the respective UbiD and CAR enzymes can be harnessed in distinct combinations to support production of a wide range of aldehydes via enzymatic CO$_2$ fixation.

**Keywords:** CO$_2$ fixation; biocatalysts; UbiD family (de)carboxylases; carboxylic acid reductases; biocatalytic cascades

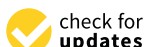



## 1. Introduction

The traditional organic chemistry synthesis required for the manufacture of pharmaceuticals and fine chemicals often relies on the application of metal-based catalysis as well as petrochemical feedstocks. These resources are often energy intensive and environmentally damaging, both in their application and their extraction [1]. The use of (waste) CO$_2$ as a building block to produce high value compounds and pharmaceutical precursors is of considerable interest, given the potential green chemistry credentials of such processes. However, the high-activation-energy barriers of C-H activation, and subsequent carboxylation, present a formidable challenge, requiring high pressures and harsh reaction conditions [2].

In contrast, biological CO$_2$ fixation offers a sustainable route to a wide range of compounds, and recent work has identified several enzyme classes capable of C-H activation and (de)carboxylation [3]. One such class is the UbiD family, a group of enzymes that rely on the modified flavin cofactor prFMN to perform the reversible (de)carboxylation of a range of aromatic and aliphatic poly-unsaturated carboxylic acids [4]. While the inherent thermodynamics usually strongly favor the decarboxylative reaction [5,6], at increased [CO$_2$] these enzymes can be used to achieve C-H activation through carboxylation at ambient conditions.

In an alternative, bioinspired approach, it has been demonstrated that by coupling this prFMN-dependent class of UbiD (de)carboxylases with carboxylic acid reductase (CAR), it is possible to overcome the inherent thermodynamic bias towards decarboxylation, even at ambient [CO$_2$]. Indeed, combining fungal Fdc1 with CAR yields carboxylation of terminal alkenes and concomitant reduction of the intermediate carboxylic acids, to produce value-added aldehydes [7].

However, Fdc1 type enzymes have limited substrate specificity compared to that seen across the collective UbiD family, with a substrate preference largely consisting of cinnamic acid derivatives, or short chain polyunsaturated aliphatic acids [5]. The varied substrate range of the UbiD family suggests that coupling distinct family members to CAR can offer a putative route to other aldehyde products. The recently characterised *Pseudomonas aeruginosa* PA0254 has been shown to catalyse the interconversion of pyrrole and pyrrole-2-carboxylic acid. With the introduction of a single point mutation, it is possible to broaden the substrate scope of PA0254 to include furan and thiophene type substrates [8].

Five-member heterocyclic aldehydes such as furfural, thiophene-, and pyrrole-2-carbaldehyde have been found to be useful building blocks in the synthesis of high value compounds such as polymers, biochemical dyes, and pharmaceuticals [9–11]. For example, pyrrole-2-carbaldehyde has been utilised in the production of the BODIPY family of fluorescent dyes [9], while furfural has applications in resin production, herbicides, and hypergolic fuels [10]. Thiophene-2-carbaldehyde has been used in the purification of palladium from aqueous solutions [11].

Hence, a system coupling PA0254 with a suitable CAR could, in principle, yield valuable heteroaromatic aldehydes and present an attractive route to this class of compounds. However, to our knowledge, no CAR enzyme has been reported to reduce pyrrole-2-carboxylic acid or the furan and thiophene counterparts [12–16].

We report that carboxylic acid reductase from *Segniliparus rotundus* (CARse) is capable of the reduction of pyrrole-, furan- and thiophene-2-carboxylic acids to their corresponding aldehydes. Furthermore, we demonstrate that coupling with PA0254 supports a one-pot biocatalytic method for the carboxylation and subsequent reduction of pyrrole to form pyrrole-2-carbaldehyde (Figure 1).

**Figure 1.** One-pot production of pyrrole-2-carbaldehyde, via a linked UbiD-CAR system, utilising whole-cell biocatalysts for improved cofactor economy.

## 2. Results and Discussion

### 2.1. Screening of Carboxylic Acid Reductases for Activity towards Five-Membered Heterocyclic Carboxylic Acids

To obtain a suitable CAR with activity with pyrrole-, furan-, and thiophene-2-carboxylic acids, we tested two specific CARs. CARse was selected for its previously described activity towards heteroaromatic carboxylic acids, functionalised at the third position [16]. In addition, *Tsukamurella paurometabola* CAR (CARtp) was selected, as previously utilised in UbiD-CAR-linked reactions [7]. Both enzymes exhibited activity with all three carboxylic acid substrates, when assayed with 5 mM substrate in the presence of 10 mM each of ATP and NADPH in 100 mM Tris buffer pH 7.5 (Figure 2). The comparative activity observed with furan-2-carboxylic acid resembled that previously observed with the 3-carboxylic acid isomer [16], with CARse displaying a relative 56% increase in furfural yield compared to CARte.

However, comparative substrate specificity differed for isomers of the pyrrole and thiophene class compounds, with CARse providing an ~18% higher yield of pyrrole-2-carbaldehyde over CARtp, while ~11% less thiophene-2-carboxaldehyde was produced by CARse versus CARtp. This appears to be the reverse of previously reported data for isomers carboxylated at the third position, where CARse displayed markedly lower activity in comparison to CARtp, while providing higher levels of reduction for thiophene-3-carboxylic acid [16]. We suspect that the differing substrate specificity observed between

these two homologues is due to minor amino acid changes in the enzyme active site or substrate access tunnel, however, additional investigation may clarify this further.

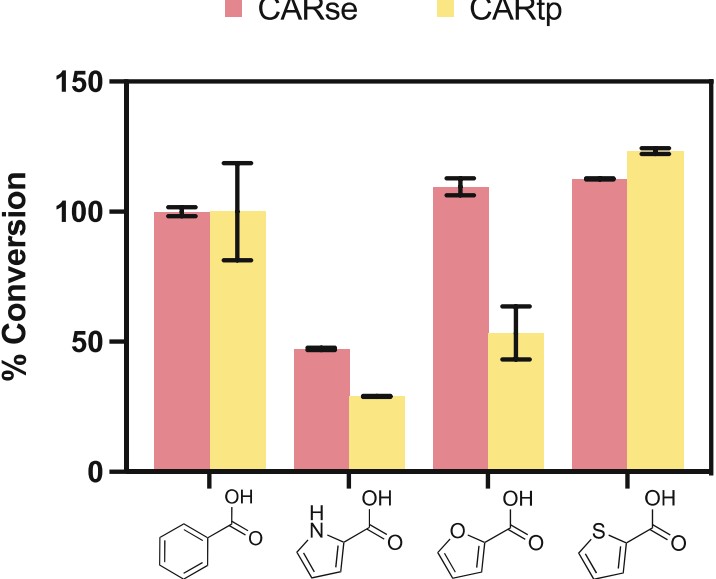

**Figure 2.** Relative percent conversion of CARse and CARtp against pyrrole-, furan-, and thiophene-2-carboxylic acids normalised against reduction of benzoic acid. Benzoic acid conversion ($4.21 \pm 0.06$ mM and $2.88 \pm 0.05$ mM for CARse and CARtp, respectively) normalised at 100%. Performed at a 500 µL scale in triplicate and analysed via HPLC, following 18 h of incubation at 30 °C.

### 2.2. Coupling of Purified PA0254 and CARse

To test the initial feasibility of the proposed one-pot carboxylation of heteroaromatics, we combined purified PA0254 and CARse and performed a pyrrole carboxylation and reduction reaction at a 500 µL scale in the presence of 2x excess of ATP and NADPH cofactors and 500 mM $KHCO_3$. This resulted in an average production of $0.89 \pm 0.10$ mM of pyrrole-2-carbaldehyde from 10 mM pyrrole substrate, equating to, approximately, 9% conversion.

Following the initial successful production of our target aldehyde, we performed a series of iterative optimisation assays, with the aim of improved yield and lowered reaction cost. Our initial target was the reliance of CARse on the reduced nicotinamide NADPH cofactor, as this poses the largest reagent cost factor within this system. As such, we tested the introduction of a nicotinamide recycling system, using glucose dehydrogenase (GDH) to reduce NADP+ at the expense of glucose. This showed that a comparable average product yield of $0.93 \pm 0.11$ mM could be obtained via supplementation of 0.25 mM NADPH in the presence of a commercially available GDH enzyme obtained from Codexis Inc. (Figure 3). In comparison, similar trials using a wildtype GDH from *Bacillus subtilis* only yielded just $0.60 \pm 0.11$ mM of product.

We next focused the optimization of the supplemented bicarbonate, as prior work has shown that the optimum bicarbonate concentration is highly specific to the carboxylase enzyme in use [3]. We found that altering the concentration of $KHCO_3$ improved the previously obtained yield by a factor of ~2, equating to $1.90 \pm 0.15$ mM of the target aldehyde at 300 mM of bicarbonate salt (Figure 4).

We surmise that the decreased activity at an even higher bicarbonate concentration may be due to an inhibitory effect on CARse-mediated catalysis or by the accompanying pH change resultant from the addition. Finally, the introduction of an ATP recycling system derived from *Acinetobacter johnsonii* [17], later trailed to further reduce the reliance of the reaction on supplemented cofactors, resulted in a 9% reduction in aldehyde yield (Figure 5).

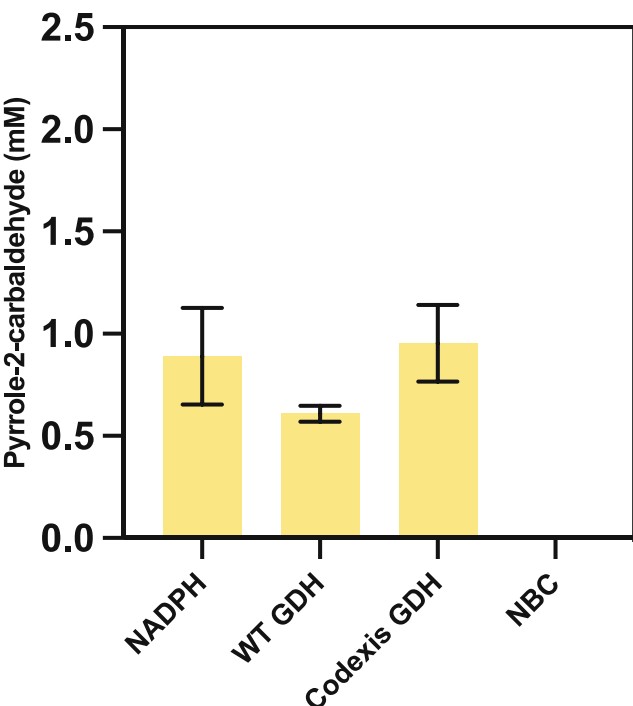

**Figure 3.** Comparative pyrrole-2-carbaldehyde yield, obtained from coupling purified PA0254 and CARse enzymes in the presence of 500 mM $KHCO_3$, as analysed via HPLC from replacement of 20 mM NADPH with wildtype bsGDH or commercially available CODEXIS CDX901-GDH at 0.30 mg/mL in the presence of 50 mM glucose. Reactions were incubated at 30 °C for 18 h. No observable carboxylation occurred in the no-biocatalyst controls (NBC).

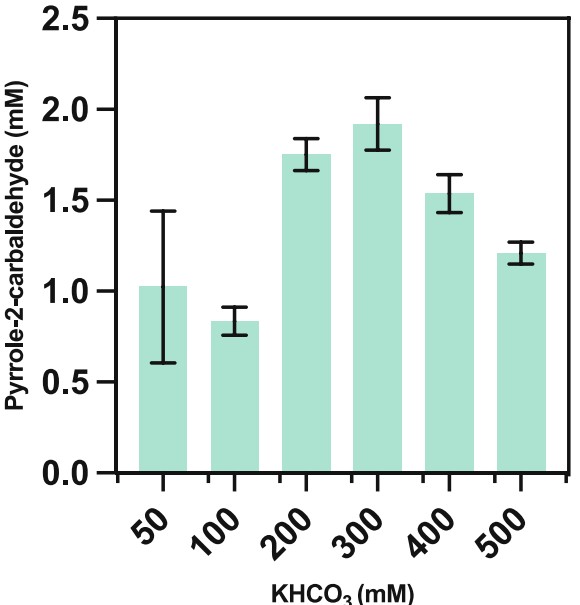

**Figure 4.** Effect of bicarbonate salt concentration on yield of pyrrole-2-carbaldehyde from 10 mM pyrrole, at conditions ranging between 50–500 mM $KHCO_3$ utilising purified PA0254 and CARse biocatalysts. No carboxylation of pyrrole was observed in controls lacking biocatalyst at any concentration of bicarbonate.

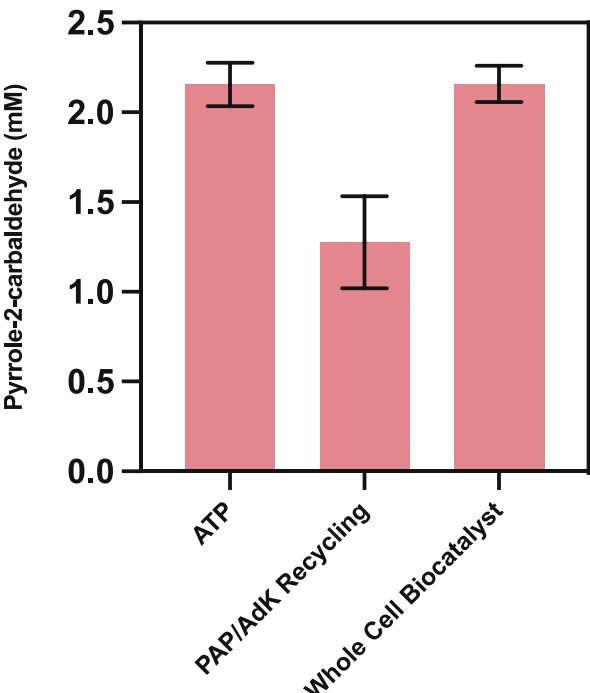

**Figure 5.** Comparative aldehyde yield obtained by the reduction of supplemented ATP from 20 mM to 0.25 mM in the presence of 0.25 mg/mL each of AjPAP and AjAdK enzymes and 4 mg/mL sodium hexametaphosphate using purified PA0254 and CARse enzymes. Compared against the yield obtained with whole-cell CARse biocatalyst at OD600 = 20, supplemented with glucose to 50 mM in the presence of purified PA0254. No carboxylation of pyrrole was observed in controls lacking biocatalysts for any condition.

### 2.3. Using Whole-Cell CARse Biocatalysts

While the introduction of a nicotinamide recycling system enabled a significant reduction in reaction cost, the <20% average yield of aldehyde obtained with a purified biocatalyst system limits the cost effectiveness of this approach, due to the expense incurred to produce the purified enzyme. Therefore, as an alternative approach, we trailed the replacement of purified CARse biocatalyst with whole *E. coli* cells, expressing both CARse and the phosphopantetheine-transferase BsSfp. This not only served to eliminate the requirement for both supplemented NADPH and ATP cofactors and their associated recycling systems, but also resulted in a comparable yield of $2.10 \pm 0.17$ mM of pyrrole-2-carbaldehyde after 18 h (Figure 5). The implementation of whole-cell biocatalyst also reduces the complexity and manufacturing costs associated with the used of purified CARse enzyme.

### 2.4. Pyrrole-2-carbaldehyde Is Unstable

Further attempts to optimise the production of pyrrole-2-carbaldehyde, via the increase in final whole-cell biocatalyst concentration from OD600 = 20 to OD600 = 50, resulted in a reduced observable product yield. Simultaneous supplementation of glycerol, as a cheap carbon feedstock for whole-cell biocatalysts, to final concentrations between 5 and 15% *v/v* further resulted in decreased aldehyde yield (Figure 6).

This may be in line with recent observations that increased buffer viscosity reduces product yield in UbiD enzymes, via inhibition of domain motion [18]. A short-time course analysis (Figure 7) indicated that the biocatalysts remained active following 24 h of incubation, so were unlikely to be undergoing significant degradation during the course of the reaction; we surmised that the reactive aldehyde product may be degrading in the presence of our biocatalyst. To test this hypothesis, we compared the peak area response of a range of freshly prepared pyrrole-2-carboxaldehyde standards (as measured on HPLC at 290 nm)

to equivalent concentrations incubated in under reaction conditions with and without the presence of biocatalyst over 18 h (Figure 8).

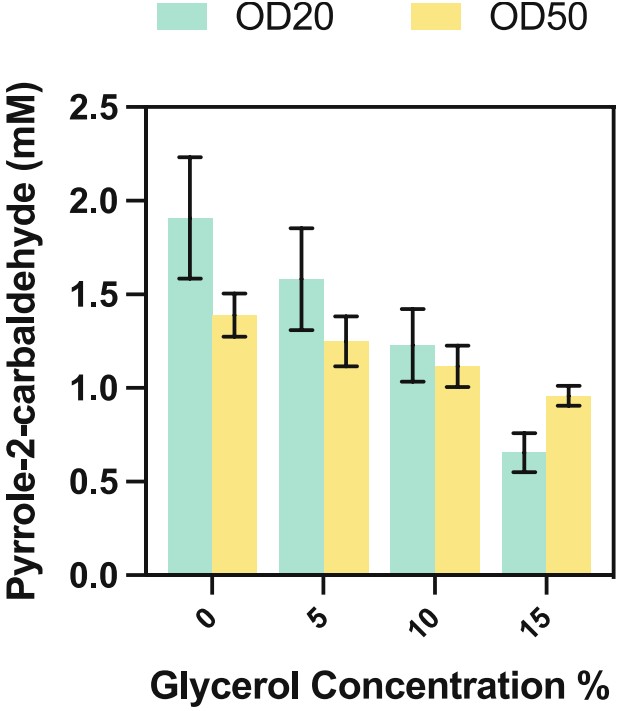

**Figure 6.** Pyrrole-2-carbaldehyde yield obtained from 10 mM pyrrole by increasing final CARse whole-cell biocatalyst concentration from OD600 = 20 to OD600 = 50, and from supplementation of glycerol in 5% *v/v* increments in the presence of purified PA0254. No product formation was observed in biocatalyst-free controls at any condition.

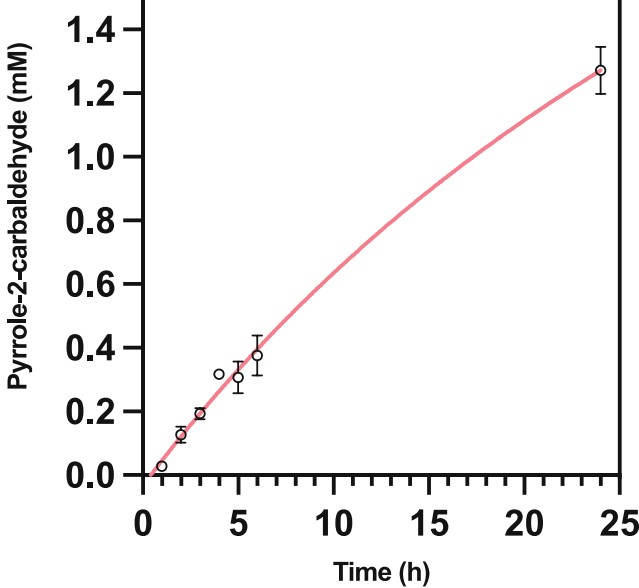

**Figure 7.** Time course analysis of pyrrole-2-carbaldehyde production from whole-cell CARse and Purified PA0254 supplied with 10 mM pyrrole, performed over a period of 6 h. Comparative data points from 24 h of incubation displayed for clarity. No carboxylation of pyrrole was observed in controls lacking biocatalyst at any time point.

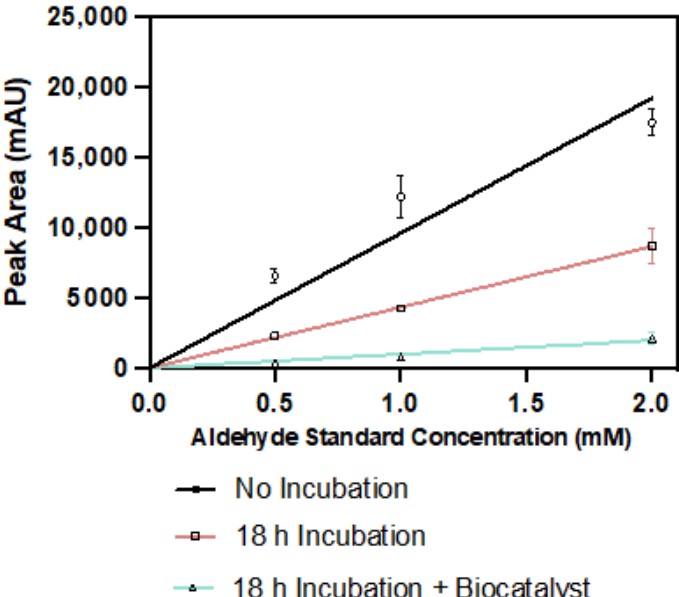

**Figure 8.** Peak area response of varying concentrations of pyrrole-2-carbaldehyde prepared fresh in 100 mM KPi pH 6.0, 300 mM KHCO$_3$, 50 mM Glucose, and 40 mM MgCl$_2$, compared to that obtained when incubated in the same conditions for 18 h at 30 °C and when incubated for 18 h at 30 °C in the presence of 0.27 mg/mL purified PA0254 and OD600 = 20 whole-cell CARse biocatalysts. Measured via HPLC at 290 nm.

We observed an average two-fold decrease in the peak area response versus the freshly prepared standard solutions, when pyrrole-2-carbaldehyde was incubated in solution for 18 h at 30 °C, which dropped, further, to a 90% reduction of the peak area when incubated for the same period in the presence of a biocatalyst. Given the inherent reactivity of aldehydes, this indicates the desired product is likely converted to other species (due to condensation and/or redox reactions) over the course of the reaction. Previous work has noted that in the presence of whole-cell biocatalysts, aldehyde products may be further reduced to form the corresponding alcohol [7,19,20], perhaps contributing to the reduction in product we observe here, so this possibility is a subject for potential further study. Hence, the present observed maximal product yield of 2.14 ± 0.16 mM likely is indicative of significantly higher production levels.

### 2.5. Neither Furan or Thiophene Class Substrates Yield Product

Prior work has shown that a point mutation of N318C in PA0254 broadens the substrate specificity to include furan- and thiophene-2-carboxylic acids, so we tested whether a CARse-linked reaction using N318C variant would act on these substrates. Given that the low boiling point of furan (approx. 31 °C) renders it an unattractive substrate for our reaction conditions, 3-methylfuran was utilised in these trials, as PA0254 can tolerate a 3-methyl substitution [8]. Unfortunately, the active (Figure 9a) N318C enzyme combined with our optimised whole-cell CARse system yielded no production of thiophene-2-carbaldehyde (Figure 9b) or 3-methylfuran-2-carbaldehyde (Figure 9c).

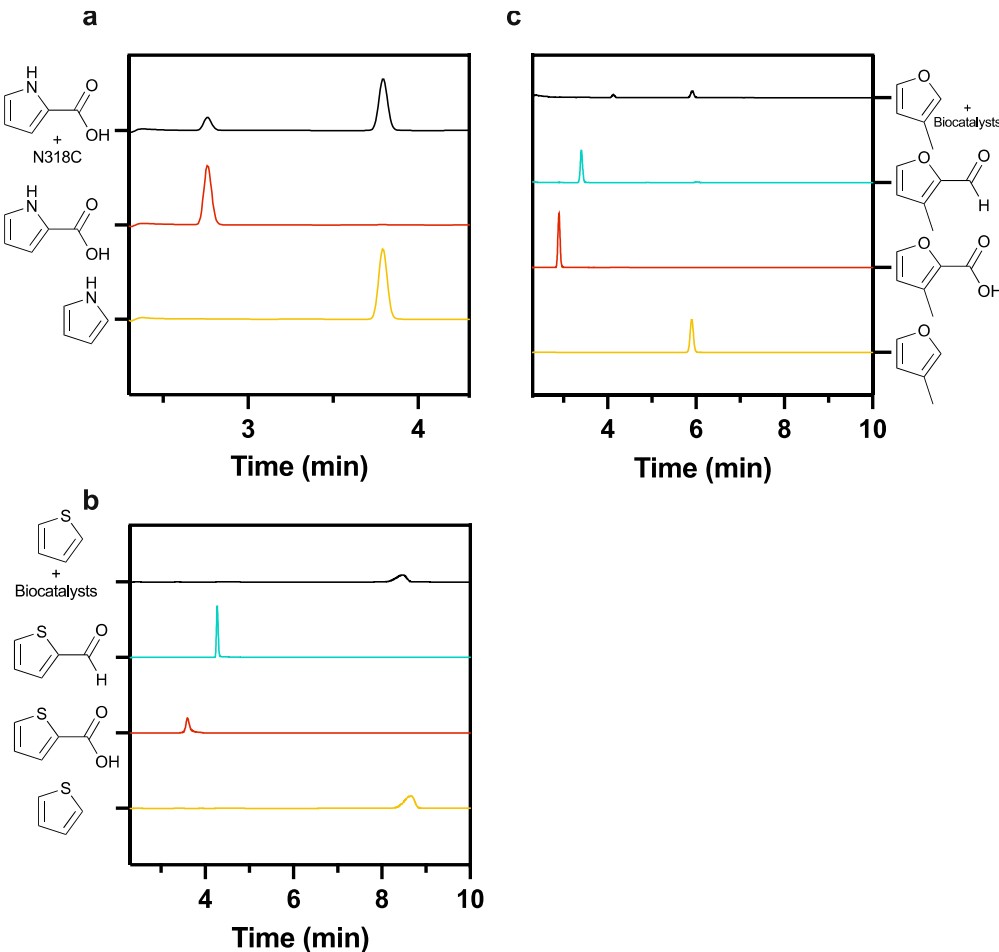

**Figure 9.** Comparative HPLC trace analysis of biocatalytic reactions utilising purified PA0254-N318C mutant against known compound standards. Incubation of 0.27 mg/mL of N318C with 1 mM pyrrole-2-carboxylic acid yields the formation of pyrrole following incubation at 30 °C (**a**). However, no aldehyde is observed upon the application of purified N318C and whole-cell CARse biocatalysts with either thiophene (**b**) or 3-methylfuran (**c**) as starting substrates.

## 3. Materials and Methods

### 3.1. Production and Preparation of Active PA0254 Biocatalysts

Heterologous expression and purification of PA0254 was performed as described previously [8]. Briefly, PA0254 was co-transformed with Pseudomonas aeruginosa UbiX in HMS174(DE3) chemically competent *E. coli* (Merck) and expressed in liquid culture via IPTG-mediated induction. The obtained cells were lysed via constant flow cell-disruption, and the active PA0254UbiX was purified from the obtained lysate via nickel-affinity chromatography performed using gravity flow. The purified biocatalyst was desalted to remove imidazole and flash frozen prior to storage at −80 °C. SDS-Page characterisation of purified biocatalysts may be found in Figure S1.

### 3.2. Production and Preparation of CARse Biocatalysts

3.2.1. Cloning and Expression

Cloning and expression of CARse was performed, as previously described [16]. In short, CARse was co-expressed in HMS174(DE3) (Merck), with a sfp-type 4′-phosphopantet-heinyl transferase from *Bacillus subtilis* in the same manner described for PA0254.

### 3.2.2. Preparation of Whole-Cell Biocatalyst

Harvested cells were washed by resuspension in sterile chilled phosphate buffered saline (PBS) solution, via gently repeated aspiration with a 50 mL serological pipette, and were transferred to 50 mL conical centrifuge tubes. Suspension was then centrifuged at $3500 \times g$ for 15 min, and the supernatant was discarded. This wash step was repeated two further times. The final cell suspension was split into 1 mL aliquots in 1.5 mL microcentrifuge tubes, and cells were harvested via a final centrifugation step and the supernatant discarded. Whole-cell biocatalyst was then either immediately frozen at $-20\,^\circ$C for storage or resuspended in 800 μL of chilled (PBS).

### 3.2.3. Preparation of Purified Biocatalyst

Harvested cells were resuspended at a 1:4 (wet cell mass:buffer) ratio in 50 mM Tris pH 7.0, 300 mM NaCl, 10 mM imidazole, and 1 mM $MgCl_2$, supplemented with complete EDTA-free protease inhibitors (Roche), DNase, and RNase (Sigma). Cells were lysed via constant flow cell disruption at 10 kPSi and 20 kPsi, sequentially. Cell lysate was centrifuged at $20,000 \times g$ for 1 h, and the supernatant filter was with a 0.45 μm filter, before application to a gravity flow column containing Ni-NTA (Qiagen) equilibrated in resuspension buffer. Following loading, the column was washed with 10 column volumes of resuspension buffer, followed by 10 column volumes of 50 mM Tris pH 7.0, 300 mM NaCl, and 40 mM imidazole. The desired protein was eluted from the column in the above buffer supplemented with imidazole to 250 mM, analysed for purity via SDS-PAGE, and desalted into 50 mM Tris pH 7.0 and 200 mM NaCl, using a CentriPure P100 gel filtration column (empBiotech). SDS-Page characterisation of purified biocatalysts may be found in Figure S1.

### 3.3. Carboxylic Acid Reductase Enzyme Screening

Qualitive screening of CARse against heteroaromatic substrates was performed using purified enzyme at a 500 μL scale. Reactions contained 0.25 mg purified CARse, 5 mM of carboxylic acid substrate, 10 mM each NADPH and ATP, 100 mM $MgCl_2$, and 100 mM Tris pH 7.5. Samples were incubated at $30\,^\circ$C for 18 h and analysed via HPLC for the desired aldehyde product. Screening was performed in triplicate against pyrrole-2-carboxylic acid, furan-2-carboxylic acid, thiophene-2-carboxylic acid, and benzoic acid substrates.

### 3.4. Initial PA0254-CARse Coupling Reactions

Reactions contained 100 mM Potassium phosphate buffer pH 6.0, 500 mM Potassium bicarbonate, 10 mM Pyrrole, 0.27 mg/mL and 0.50 mg/mL purified PA0254 and CARse biocatalysts, respectively, of 20 mM ATP (pH 7.0), 20 mM NADPH, and 40 mM $MgCl_2$. Final assay pH was measured at pH 7.4. Assays were performed in triplicate against a biocatalyst free control incubated at $30\,^\circ$C for 18 h.

### 3.5. Optimisation of Assay Conditions

#### 3.5.1. Nicotinamide Recycling

Comparison of nicotinamide cofactor recycling was performed as described in the initial coupling reaction, with a reduced NADPH concentration to 0.25 mM, and the addition of 20 mM glucose and 0.30 mg/mL of either purified wildtype glucose dehydrogenase from *B. subtilis* or a commercially available alternative (CDX-901-CODEXIS).

#### 3.5.2. Bicarbonate Optimization

Bicarbonate concentration assays were performed by incorporating the lowered NADPH concentration of 0.25 mM and the addition of 0.30 mg/mL CDX-901 GDH (Codexis) and 20 mM glucose to the initial coupling reaction conditions. The concentration of potassium bicarbonate was then varied in 100 mM increments between 100 mM and 500 mM, in addition to one condition at 50 mM. The final pH of each condition was measured, as shown in Table 1. All assays were performed in triplicate in comparison to biocatalyst-free controls and assayed via HPLC.

**Table 1.** Effect of concentration of $KHCO_3$ on final assay pH.

| Concentration of $KHCO_3$ (mM) | Measured pH |
|---|---|
| 50 | 6.49 |
| 100 | 6.74 |
| 200 | 7.01 |
| 300 | 7.16 |
| 400 | 7.35 |
| 500 | 7.44 |

### 3.5.3. ATP Recycling

Assays to characterise the effect of the introduction of a two-enzyme recycling system were performed at a 500 μL scale containing 100 mM potassium phosphate pH 6.0, 300 mM $KHCO_3$, 40 mM $MgCl_2$, 20 mM Glucose, 0.27 mg/mL PA0254, 0.50 mg/mL CARse, 0.3 mL CDX-901, and 10 mM pyrrole. In addition, the recycling system consisted of 4.00 mg/mL sodium hexametaphosphate, 0.5 mM ATP (pH 7.0), and 0.25 mg/mL, each purified adenylate kinase (AdK) and polyphosphate:AMP transferase (PAP) from *Acinetobacter johnsonii*. Assays were performed in triplicate vs an ATP-excess control, which replaced the recycling system with 20 mM ATP (pH 7.0). The final assay pH was measured at pH 7.22.

### 3.5.4. Whole-Cell CARse Biocatalyst Assays

The effect of the use of CARse whole-cell biocatalysts to enable economical in situ ATP and NADPH production was assessed in the following conditions at a 500 μL scale: 100 mM potassium phosphate pH 6.0, 300 mM $KHCO_3$, 50 mM Glucose, 40 mM $MgCl_2$, 10 mM Pyrrole, 0.27 mg/mL PA0254, and CAR9 whole cells at a final OD600 of 20 or 50. Glycerol supplementation was performed via the addition of 50% sterile glycerol to 5%, 10%, or 15% final volume.

### 3.5.5. Pyrrole-2-carboxaldehyde Degradation Assay

To assess the stability of pyrrole-2-carboxaldehyde over time, freshly prepared 0.5, 1.0, and 2.0 mM stocks of the aldehyde were prepared in 100 mM KPi pH 6.0, 300 mM $KHCO_3$, 50 mM Glucose, and 40 mM $MgCl_2$, and the peak response area was determined via HPLC. Identical stocks were, concurrently, prepared in the same buffer and incubated at 30 °C for 18 h, both with and without the addition of 0.27 mg/mL purified PA0254 and whole-cell CARse biocatalyst, to a final OD600 of 20. Peak area response was then determined via HPLC, as previously performed. All conditions were prepared in triplicate.

### 3.6. Analytical Procedures

All samples were analysed on an Agilent 1260 Infinity HPLC equipped with an Ascentis C18 581325-U (Supelco) or a Kinetex 00G-4601-E0 column (Phenomenex). The mobile phase consisted of varying ratios of water and acetonitrile modified with 0.1% *v/v* TFA, as described below. The quantification of aldehyde products was assessed via comparison to peak area response of analytical standard at varying concentration.

PA0254-CAR coupling reactions utilising pyrrole were extracted via the addition of 500 μL of MTBE, brief vortexing, incubation at 30 °C for 5 min, and centrifugation at 15,000× *g* for 30 min at 4 °C. The organic layer was then removed and analysed via HPLC, utilising a 50:50 ACN:$H_2O$ isocratic method for 6 min at a 1 mL/min flow rate monitored at 290 nm.

Reactions utilising thiophene and 3-methylfuran substrates were quenched via the addition of 1 reaction volume of acetonitrile supplemented with 0.1% *v/v* trifluoracetic acid. Samples were then incubated at 30 °C for 5 min, before centrifugation at 15,000× *g* for 30 min at 4 °C. Thiophene samples were analysed using a 10 min 60:40 ACN:$H_2O$ isocratic

method and monitored at 240 nm, while 3-methylfuran was analysed at 50:50 ACN:H$_2$O over the same time period and monitored at 220 nm.

## 4. Conclusions

We conclude that an optimised PA0254-CARse one-pot reaction system is capable of carboxylation of pyrrole to pyrrole-2-carbaldehyde, but it suffers from low yield due to the instability of the target product. Similar observations of aldehyde instability in biocatalytic systems have been made, when coupling CAR enzymes with the prFMN-dependent indole-3-decarboxylase from *Arthrobacter nicotianae* [21], which noted the degredation of indole-3-carboxaldehyde product following extended incubaiton. This further highlights the applicability of a linked UbiD-CAR system, though future work may improve on this system in the stabilization or scavenging of the aldehyde, either via enzymatic means such as linkage to an imine reductase for the production of allylic amines, as previously described in the further modification of UbiD-CAR-produced cinnamaldehyde with *Cystobacter ferrugineus* imine reductase to produce *N*-cinnamylcyclopropylamine [7], or via methods of in situ product recovery such as biphasic reactions [22]. However, this work provides proof of principle of an efficient biosynthetic toolkit to produce heteroaromatic aldehydes and may be applied in future in the biological synthesis of their derivatives (Figure 10). Furthermore, it demonstrates that the relatively broad natural substrate specificity of the UbiD, and CAR enzyme families can be harnessed to deliver a route to a range of aldehydes, by tailoring the enzyme combination to ensure efficient CO$_2$ fixation.

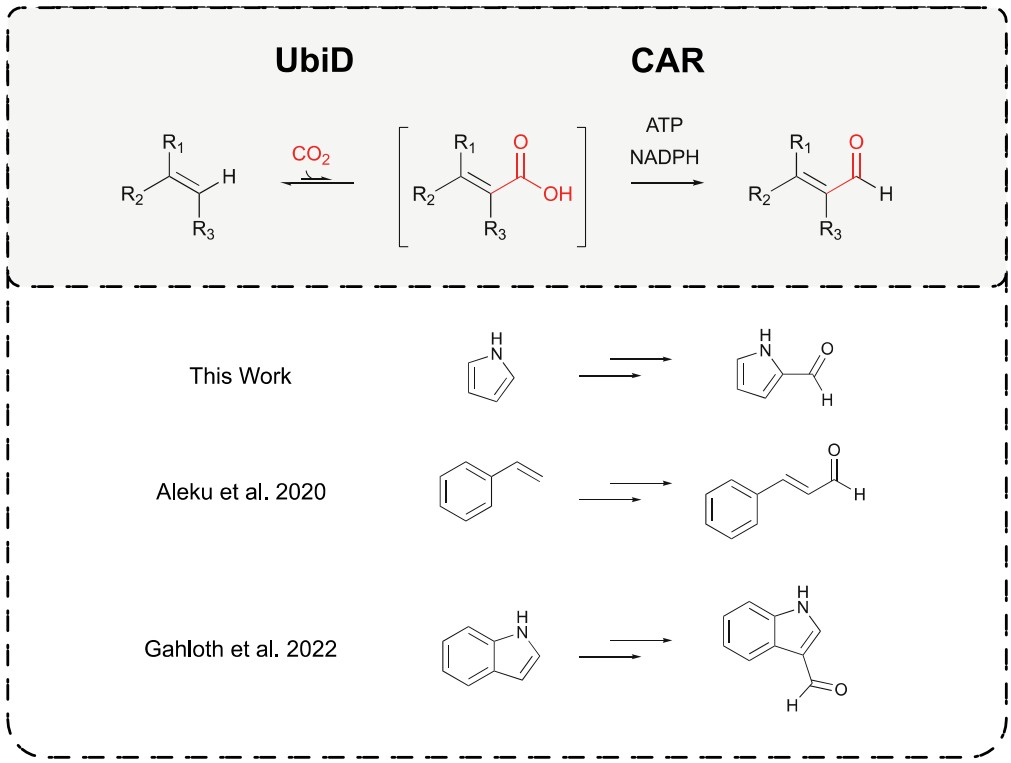

**Figure 10.** Application of combined UbiD-CAR biocatalytic reactions for the production of aldehydes from terminal alkanes and heteroaromatics. Highlighting successes to date in utilising this system in the production of pyrrole-2-carbaldehyde from pyrrole, as presented within this work, the production of cinnamaldehyde from styrene [7], and the production of indole-3-carbaldehyde from indole [21].

**Supplementary Materials:** The following supporting information can be downloaded at: https://www.mdpi.com/article/10.3390/catal12050538/s1, Figure S1: SDS-Page analysis of purified biocatalysts.

**Author Contributions:** Conceptualization, G.R.T. and D.L.; methodology, G.R.T.; investigation, G.R.T., S.A.M. and H.M.; formal data analysis, G.R.T.; resources, D.L.; writing—original draft preparation, G.R.T.; writing—review and editing, D.L.; visualization, G.R.T.; supervision, D.L.; project administration, G.R.T. and D.L.; funding acquisition, G.R.T. and D.L. All authors have read and agreed to the published version of the manuscript.

**Funding:** This research was funded by the Biotechnology and Biological Sciences Research Council (BBSRC), grant numbers BB/R01034X/1 (D.L.) and BB/M011208/1 (D.L. and G.R.T.), and by the European Research Council (ERC) grant pre-FAB ADG_695013 (D.L.). The APC was funded by the ERC grant pre-FAB ADG_695013.

**Data Availability Statement:** The data presented in this study are available on request from the corresponding author.

**Conflicts of Interest:** The authors declare no conflict of interest.

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
