# Peer review of "Biosynthesis of Pyrrole-2-carbaldehyde via Enzymatic CO2 Fixation"

_catalysts, doi:10.3390/catal12050538_

Round 1

Reviewer 1 Report

The work "Production of Pyrrole-2-carbaldehyde via Enzymatic CO2 Fixation" by a team of scientists from the United Kingdom is devoted to the description of the original biocatalytic one-pot carboxylation reaction (with subsequent reduction). SAR acted as an enzyme in the described reaction of the conversion of pyrrole to pyrrole-2-carbaldehyde, and the carboxylation process was carried out with the fixation of carbon dioxide, which is of great importance from the point of view of green chemistry. The work was done and presented at a high level. The authors clearly indicate the relevance and importance of the described study. The paper presents data on the screening of SARs in relation to activity to 5-membered aromatic cycles. A separate advantage is the study of the effectiveness of SARs in relation to isomeric forms of a number of heterocycles. The authors have shown that the described reaction of pyrrole carboxylation to pyrrole-2-carbaldehyde works, but has a low yield of the target product. Nevertheless, the very fact of such a reaction is important and serves as a basis for further studies of the reactions of obtaining heteroaromatic aldehydes using biocatalysis. There are some shortcomings in the work that should be corrected, namely, the authors should change the title of the second part to "Results and Discussion" and the title of the third part to "Conclusions". It is also worth paying attention to the dimensions and rounding of values ​​and bring everything to uniformity, for example, outputs are given both to tenths and to hundredths. After making changes, the work can be accepted for publish.

Author Response

  • ‘…the authors should change the title of the second part to "Results and Discussion" and the title of the third part to "Conclusions".

The titles of these sections have been amended as suggested.

  • It is also worth paying attention to the dimensions and rounding of values ​​and bring everything to uniformity, for example, outputs are given both to tenths and to hundredths.

Values have been reported to uniformity of precision as requested. All output values are now presented to two decimal places.

Reviewer 2 Report

The presented manuscript describes efforts towards a two step enzymatic cascade to pyrrole-2-carbaldehyde. Carboxylation is combined with acid reduction. The concept was published by the authors before and also pyrrole was shown as a UbiD substrate. The idea is smart but the study and its presentation have both some weaknesses:

  • The title is misleading. ‘Production’ is annotated with preparative/industrial scale but the paper is restricted to analytical scale reactions on the low mM range of product concentrations. ‘Formation’ or some similar expression would be more appropriate to reflect the content
  • Line 32: A sustainable route or sustainable routes
  • Figure2: ‘Relative percent activity’: Results are obtained by end-point measurements. It is conversions, not activity. What was the absolute conversion of benzoic acid?
  • Line 97: Add the respective reference.
  • Throughout the manuscript, check numbers in chemical formulae. Many are not subscript (e.g. line 103)
  • Figures 3-9: Please improve Fig captions and name in every case the decarboxylase and CAR (for the CAR also whether it was purified or not). One example: Fig4: was this only CAR or the combination? Figures also miss control reactions. Is there background carboxylation at high carbonate concentration?
  • More importantly is perhaps the fact that aldehydes are readily reduced to alcohols under reductive conditions. There is extensive literature about this detoxification mechanisms (Kunjapur 2015 10.1128/AEM.03319-14, Horvat 2020 10.1002/cctc.202000895…). The authors are aware of this (Aleku 2020, Fig2), but do not consider/discuss the possibility that aldehyde did not accumulate because it was reduced to alcohol. Also Fig 8 suggests that biocatalyst is using up the aldehyde. Conditions for experiment in Fig 8 are not described (buffer? Glycerol? Which components?)
  • Why was the decarboxylase not used as whole cell catalyst?
  • Fig7: Fig captions says: comparative yield displayed, but this isn’t there.
  • Fig9: why is a showing the reverse reaction? Could there be double peaks (Aldehyde + alcohol) in the reaction with decarboxylase/CAR + pyrrole?
  • The final pH of the reactions must be specified. The buffer capacity is too low to maintain it, so why was this buffer used at all?
  • Section 4.1: the E. coli strain used must be specified. Why were competent cells used for biotransformations?
  • Section 4.3: furane/methylfurane not mentioned.

Reviewer 3 Report

  1. Error bars should be added in all figures.
  2. Characterization of the purified biocatalysts should be included in the results.
  3. Why did activity of CARse and CARtp against pyrrole-, furan- and thiophene-2-carboxylic acids differ form each other?
  4. Instead of quoting the previous work, the authors can add more discussion into the present work and the data analysis shoud be conducted more comprehensively .
  5. I recommend the authors to revise the abstract . The use of CO2 is not a highlight in this manuscript.
  6. There are several mistakes in Engligh writing. Please check the whole manuscript.
  7. Evidence that pyrrole-2-carbaldehyde is unstable should be given. Was it transformed into other substance?
  8. The authors study the effect of bicarbonate salt concentration on yield of pyrrole-2-carbaldehyde. Were there any other effects that can influence the yield of pyrrole-2-carbaldehyde?

Author Response

  • Error bars should be added in all figures.

Figures have been updated to include error bars as suggested.

  • Characterization of the purified biocatalysts should be included in the results.

SDS-Page gels of purified PA0254WT, PA0254N318C, and CARse have been provided as supplementary figure S1. Full biophysical characterisation of these enzymes has been previously described in Finnigan et al. (Reference 14) and Duan et al. (Reference 12) and as such is outside the scope of this publication.

  • Why did activity of CARse and CARtp against pyrrole-, furan- and thiophene-2-carboxylic acids differ form each other?

Added to the discussion at line 99 the potential cause of differing substrate specificity between CARse and CARtp. However, full biophysical investigation of this difference lays outside the scope of this work.

  • Instead of quoting the previous work, the authors can add more discussion into the present work and the data analysis shoud be conducted more comprehensively .

We hope to have improved on discussion and data analysis by adding additional description as requested.

  • I recommend the authors to revise the abstract . The use of CO2 is not a highlight in this manuscript.

We feel the abstract accurately reflects our results, and presents the wider issues (in terms of the need to develop routes to use CO2) in the context of which this works needs to be assessed.

  • There are several mistakes in Engligh writing. Please check the whole manuscript.

The manuscript has been reviewed and errors in spelling, grammar, and formatting  have been corrected.

  • Evidence that pyrrole-2-carbaldehyde is unstable should be given. Was it transformed into other substance?

The manuscript has been revised to state that the provided data suggests rather than asserts, that the pyrrole-2-carbaldehyde undergoes degradation. We proposed this is reasonable, due to prior work (cited), and the inherent reactivity of aldehydes.

  • The authors study the effect of bicarbonate salt concentration on yield of pyrrole-2-carbaldehyde. Were there any other effects that can influence the yield of pyrrole-2-carbaldehyde?

As well as the effect of bicarbonate salt concentration noted by the reviewer our study also presents data the effect of cofactor regeneration technique (Figures 3 and 5), biocatalyst format (Figure 5) , whole cell biocatalyst loading, and glycerol concentration (Figure 6) on the yield of pyrrole-2-carbaldehyde in this system.